# Point Cloud-Oriented Inspection of Old Street's Sustainable Transformation from the Ceramic Industry to Cultural Tourism: A Case Study of Yingge, a Ceramic Town in Taiwan

**Naai-Jung Shih *** , **Wen-Tse Hsu and Pei-Huang Diao**

Department of Architecture, National Taiwan University of Science and Technology, 43, Section 4, Keelung Road, Taipei 106, Taiwan

\* Correspondence: shihnj@mail.ntust.edu.tw; Tel.: +886-02-2737-6718

**Abstract:** Yingge, a ceramic-producing town in Northern Taiwan, has experienced three development stages in the 50 years since 1970. The town's fabric and the second contour evolved through the transformation of its former manufacturing industry into cultural tourism on Old Street. This process of evolution is evidenced through chronological changes of overlaid sections, skylines, and horizontal sections along Old Street since 1970. The street fabric has been shaped by its historical background, government planning strategies, commercial activities, cultural identity, and living patterns. Three-dimensional (3D) scans supported our analysis by capturing and segmenting the vocabularies and overlaid sections with special characteristics and changes. Commercial spaces and open street spaces were found to be mutually influential. A flexible and sometimes hidden spatial structure of fabric was elucidated. Yingge has become a large-scale shopping mall and important window into cultural tourism, with its fabric and contours redeveloped to be consistent with the identity of nearby cities. 3D scanning data were combined with documentation and maps to create a referable connection between reality and chronological data. An augmented reality (AR) application was used to simplify the inspection process through a productive connection between as-built scans and user interactions.

**Keywords:** urban fabric; second contour; 3D laser scan; ceramic industry; cultural tourism; skyline

---

## 1. Introduction

Yingge, Taiwan, formerly named Jingde (also Jindezhen), was renowned for its ceramic industry. This industry, which originated in China during the Qing Dynasty, spread to Tuzihkend, Yingge, where local soil, wheel-thrown pottery devices, snake kilns, and fuel were used for small- or medium-scaled products. The manufacturing sites were spread to Jianshan due to its black ceramic soil.

Local industry and transportation experienced major developments and transformations during the period of Japanese rule. For example, a major railroad was installed and monopolies of expertise were forbidden, with the introduction of new industrial regulations regarding the sharing of ceramic manufacturing knowledge. Later, a new coal-fueled square kiln design was introduced, which considerably improved the product quality, enabling the creation of intricate textures.

A major structural transformation occurred to the industry and urban fabric of Yingge. The number of factories grew dramatically from 50 to 247 between 1950 and 1980 [1]. A chimney-filled skyline developed where the square kilns were located at an approximate height of two to three stories. In the late 1980s, a depression occurred when wages rose and China's market opened. Another major industry transformation started in 1990 when the government committed itself to the promotion of

ceramic culture tourism. With the renovation of Old Street in 1990 and the completion of a new ceramic museum in 2000, Yingge has gradually developed into a well-known town for tourism based on its ceramic culture.

According to the Tourism and Travel Department of the New Taipei City Government, Yingge District Office [2], and local tour guides, Old Street of the Yingge district consists of five main streets (Figure 1). The streets are popular tourist locations (Figure 2), being replete with ceramic-related merchandise, do-it-yourself (DIY) facilities, and exhibitions.

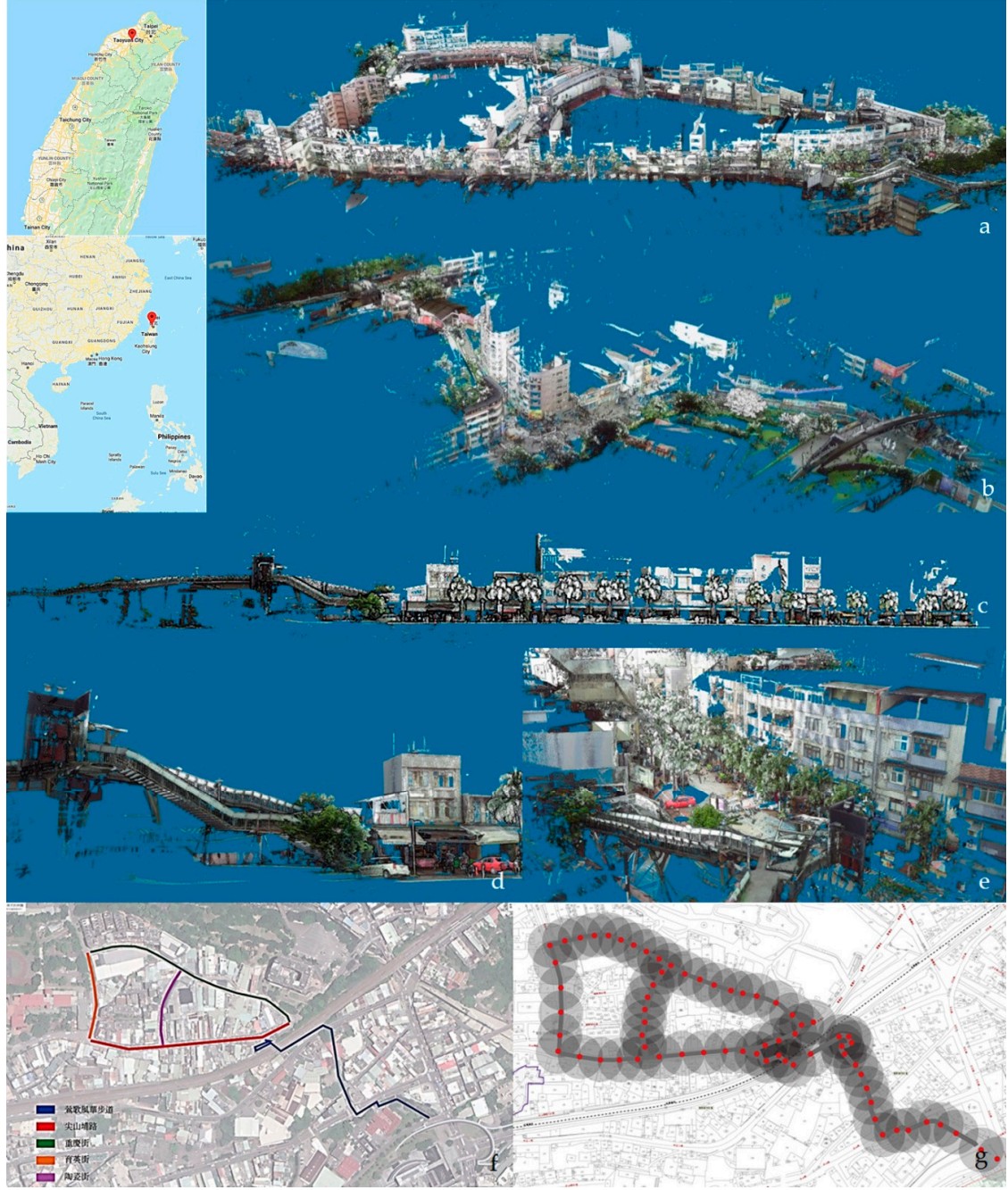

**Figure 1.** (**a**) Old Street, (**b**) the connection between the ceramic museum (lower right corner) and the entrance to Old Street, (**c**–**e**) the relationship between the skywalk and Old Street, (**f**) an aerial image of Old Street, and (**g**) the locations of scans along the street.

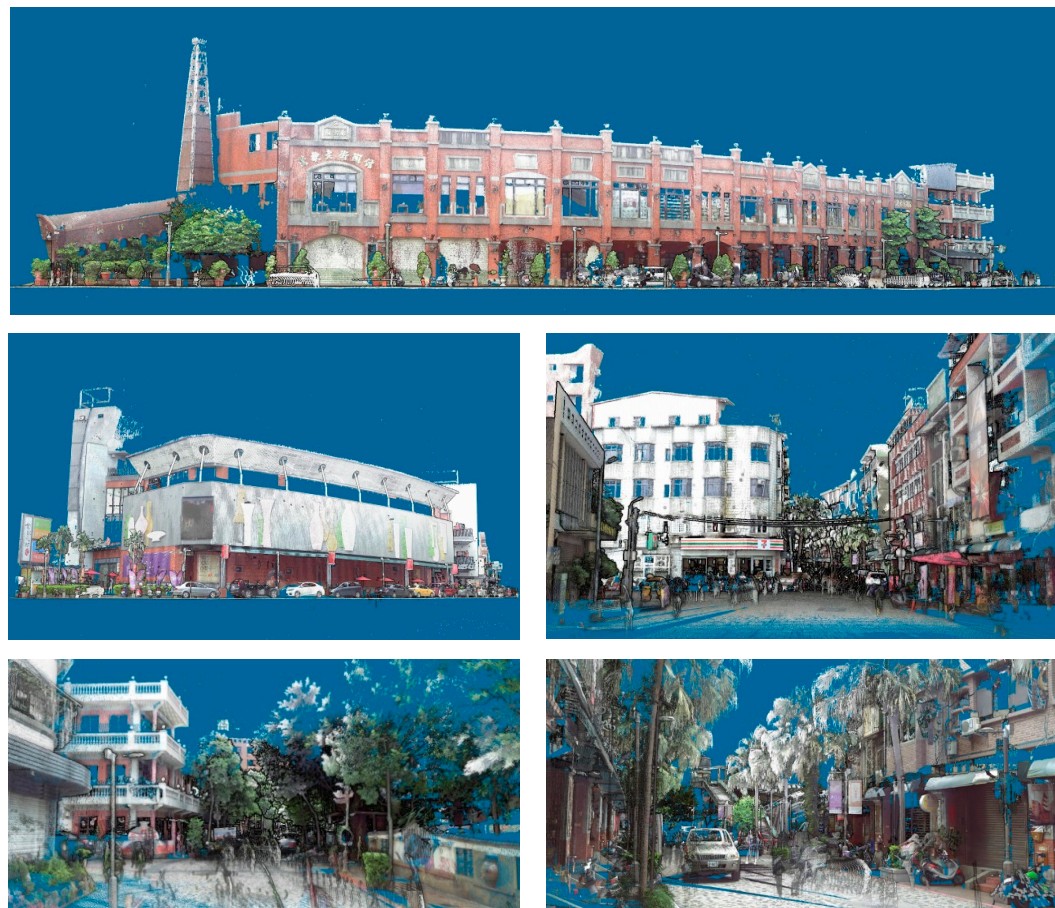

**Figure 2.** Iconic street scenes in the point cloud model.

## 2. Related Work

The transformation of industrial structures and urban environments is closely correlated with the environment of a city [3]. The transformation represents the rise, prosperity, recession, and evolution of local industry, while exerting a persistent impact on the city's urban fabric. With the aim of preserving historical heritage and Old Street, cultural tourism has evolved and is associated with sustainable strategies to redevelop unused space. Redevelopments create new meanings for old buildings, integrate local industrial characteristics, and create identities and responsibilities for local people and organizations. The heritage and Old Streets also require continual maintenance. In Yingge, the transformation changed the local urban fabric and second contour from an old manufacturing industry to cultural ceramic tourism. A reputable commercial district has been created with unique characteristics for the sustainable development of local culture and tourism.

Sustainable development in policy-making has profound implications for the practice and politics of urban and regional development [4]. A sustainable city can improve satisfaction with city life by offering many positive experiences and perspectives [5–7]. Culture and heritage are now widely perceived to be important contributors to a sustainable development [8]. The economic, environmental, and social components of sustainable development are interdependent and mutually reinforced [9]. Based on the historical changes of urban fabric around Old Street from 1970 to 2017, significant development on Old Street has been constantly occurring as part of the economic development. Considering the interdependence of the three sustainable components, economic development has to be connected with social development and environmental protection as well. The level of interdependence and mutual reinforcement, which may vary between cities, provides a trail of local history.

The Japanese architect, Ashihara Yoshinobu, defined the second contour as objects and temporary installations protruded from building enclosures, in contrast to those of the original state and appearance, which are the first contour [10]. Yingge's Old Street is full of commercial activities and residential needs that contribute to the creation of the second contour and the characteristic street space. The differences have also extended to temporary installations in festivals, as well as the space layout during both business and non-business hours. All the modifications of fabric and second contour represent an attempt to sustainably evolve the culture-related spatial structure, developed and managed by local efforts and government promotions.

The development of skylines can be related to the changing urban fabric using building information noted on cadastral maps. A skyline is a unique pattern of a city, from which mentalities, diversity, and cultures are perceived [11]. This uniqueness expresses an abstraction of a city's image and identity in terms of its spatial, historical, social, cultural, and economic structures over time [12]. The transformation of an industrial city, which is part of sustainable development, is reflected by the profile of a skyline [13]. A skyline can be considered a collection of entities that have evolved and can be used to chronologically examine a city's path of sustainability. These changes can also be found in Yingge, where the transformation included profile changes from low-rise bungalows to multi-layered skylines with characteristic cultural and commercial images.

Three-dimensional (3D) scans have been used for data collection in large areas to create urban models. The point cloud can be used to analyze urban open space, buildings, and landscapes [14–18]. The as-built information at full-scale provides a high level of precision for measurement with detailed components. In contrast to drawings or maps that are not currently updated, 3D scans provide an efficient and effective on-site data collection approach for 3D configurations that are usually unavailable.

Augmented reality (AR) has been widely applied in recent years. It enables real and virtual information in a real environment to be combined, registered, and interacted with in real-time [19–21]. AR applications have been successfully implemented in navigation, education, industry, and medical practice [22–26]. In tourism, AR can provide tourists with detailed local information [27–30]. A user can explore reality with new layers of information using mobile AR applications for a new, interactive, and highly dynamic experience [31,32]. With a geological view of relics and associated landscapes, social awareness of culture can also be enhanced, and geological heritage can be protected [33].

## 3. Research Goal

The goal of this study was to inspect the transformation of a ceramic-producing town in Taiwan, Yingge, from ceramic industry to cultural tourism, using the 3D scans of Old Street as-built scenes. The point-cloud-oriented spatial analysis of Old Street should facilitate the illustration of the cultural character and space experience of tourists. The analysis needs to present the structural composition and symbolic expressions of façades, the characteristic vocabularies of street, the second contour types and their distribution, the distribution of shop types and display patterns, the possible hidden fabric between business and non-business hours, and a substantiated perception of skylines.

The urban fabric and second contour evidence a city's development. The composition of both can verify the details of demolition and new construction according to the stages promoted by the government or local residents. A sustainable development process is correlated with today's fabric, often through difficulties that the town has experienced. As Yingge has undergone three stages of development, it is necessary to explore achievement based on interrelationships between the as-built scenes and old maps or documentation.

Point clouds, which are usually considered to be as-built descriptions of an environment, actually feature a representation system that links the past and the present. The data function as a spatial framework reference in terms of elements, patterns, forms, and changes. This major advantage should be productively applied to interpret the development history of Yingge. The feasibility of correlating a comparison between the past and present should be verified. An illustration and solid quantitative estimation should be provided of the changes that have occurred.

## 4. The Three Development Stages of Old Street

The transformation of Yingge urban fabric occurred in three main stages: (1) The major reconstructions of street blocks and skylines (1970–1997), (2) a reputable commercial district created by the implementation and development of government projects (1997–2005), and (3) new constructions (2005 and after).

### 4.1. Major Reconstruction of Street Blocks and Skylines (1970–1997)

The ceramic industry technology in Yingge changed in the late 1960s due to a new coal-burning prohibition policy and the adoption of new kiln technology. The government provided loans that were free of interest for the replacement of old square kilns with gas kilns that were smaller, more efficient, and produced less air pollution. Chimneys were also torn down due to safety concerns. These modifications removed the skyline formed by towering chimneys and the former square kilns under the chimneys. A new plan, which was initiated between 1960 and 1994, was introduced to demolish old buildings, reconstruct street blocks with straighter boundaries, and broaden the street. A chronological comparison of the cadastral maps, with properties marked by floor numbers, reveals the variations in the skyline across these decades.

### 4.2. The Redevelopment of the Reputable Commercial District (1997–2005)

In 1996, the Department of Commerce, Ministry of Economic Affairs, initiated the Shopping Street Development Promotion Project and the Reputable Commercial District Shaping Project to promote local businesses through improvement of the commercial environment. New renovations were completed in the pedestrian walking area, with new pavement, building styles, street furniture, lighting facilities, navigation systems, landscapes, fixed and moveable vehicle stoppers, manhole covers, and shop signage. In addition to the straightened street blocks, more detailed enhancements were completed in this area with developed facilities and installations that contributed to the appearance and the second contour.

### 4.3. New Construction Period (2005–Now)

Many popular attractions that have been added to Old Street since 2005 (Figure 3) have redefined street scenes. Satellite images show that new shopping malls were added to the street and contributed to the changes in the skyline. Major attractions include a new skywalk that connects Old Street and the new ceramic museum, a new shopping mall that combines the cultural and creative industry and exhibitions, and a new coffee shop chain that integrates interior decoration with Yingge ceramic characteristics. Although the scale of individual attractions is limited, consumption patterns have changed substantially. The attractions also form part of the urban fabric and second contour, which features more straightforward connections to local culture, especially as revealed by the scanned data.

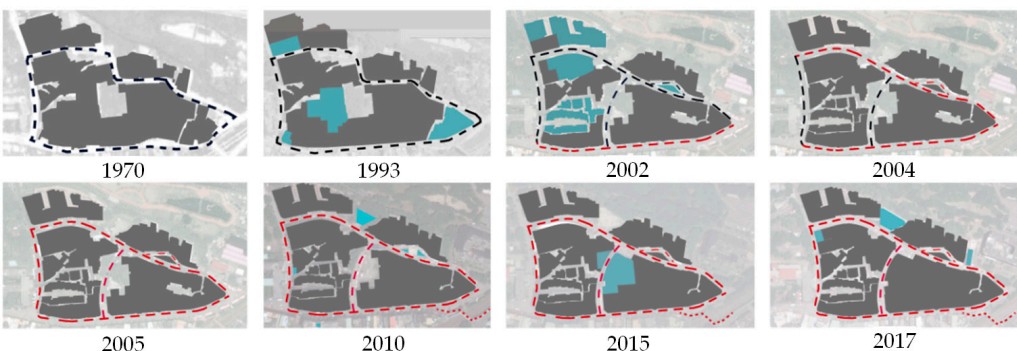

**Figure 3.** Historical changes in the urban fabric around Old Street from 1970 to 2017.

Since the challenge created by the transformation of the ceramic industry that occurred in the 1990s, tourism of ceramic culture has been actively promoted by the government. The second contour has changed due to government promotion projects (Figure 4) as follows [34]: (1) 1997: designation and construction of pedestrian walking areas; (2) 1999: First stage of detailed design for the pedestrian walking area was completed to improve pavement, architectural style, street furniture, navigation systems, landscape, and lighting facilities; (3) 2004: Third stage of construction was completed to improve pavement, lighting facilities, and fixed moveable vehicle stoppers; (4) 2005: Fourth stage construction was completed to unify shop front signage, moveable vehicle stoppers, and manhole covers; and (5) 2009: An additional stage of the pedestrian walking area project was completed to reconstruct the skywalk that connects Old Street and the new ceramic museum.

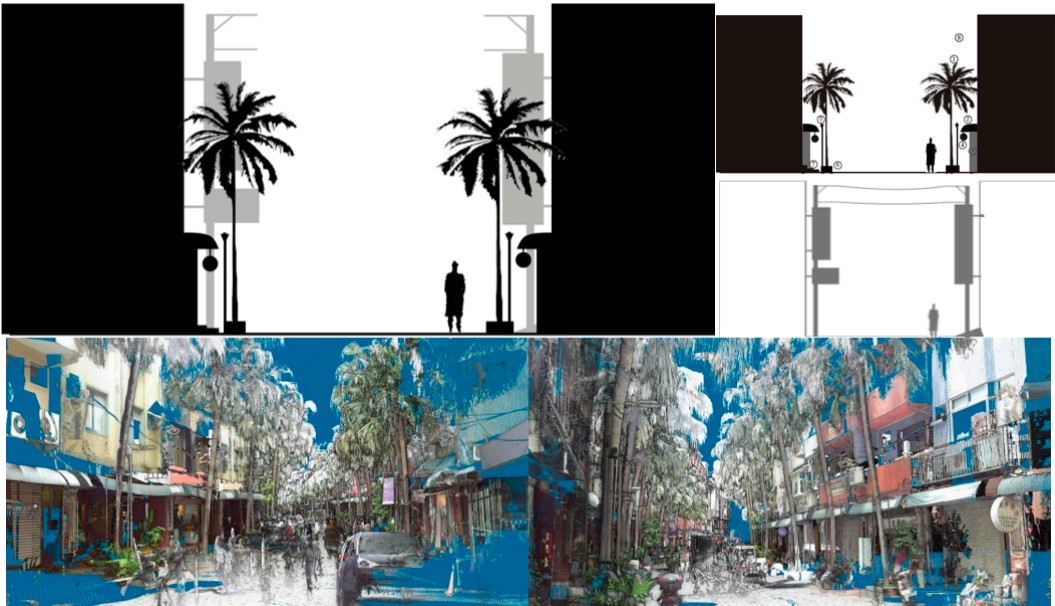

**Figure 4.** The new second contour (in black) and the former one (in gray) (above), and current as-scanned Old Street scenes (bottom).

## 5. 3D Scans of Old Street

Scans were recorded of Old Street and the connecting paths in 68 scan locations, for a total of 15.7 GB and 1,072,398,483 points, in a range approximately 320 m wide and 150 m long (Figures 1, 2 and 4). The elevated terrain and crowds of tourists contributed a certain degree of difficulty to the scanning.

Compared to photogrammetric modeling achieved by unmanned aerial vehicle (UAV) or unmanned aircraft system (UAS), a mid-range 3D scanner, Faro Focus 3D® (FARO Technologies Inc., Stuttgart, Germany), was used to capture wires, light poles, landscapes, installations on building façades, etc., with detailed geometries that were sufficiently clear to facilitate inspection of the second contour. A point cloud model, which we were able to use to identify subjects that were closely related to the second contour, was also able to create a mesh model for AR user interaction.

The point cloud model was manipulated with CloudCompare® (EDF R&D, Paris, French), Geomagic Studio® (former Raindrop Geomagic Inc., North Carolina, USA, now 3D Systems Company, State of South Carolina, USA), Meshlab®(Visual Computing Lab – ISTI – CNR, Pisa, Italy), and Autodesk Revit® software (Autodesk Inc., San Rafael, California, USA) to retrieve sections, skylines, ground floor plans, and images or vector drawings for analysis.

## 6. A Point Cloud-Oriented Spatial Analysis of Old Street's Cultural Character and Experience

The combination of the ceramic town and Reputable Commercial District created distinguishing cultural characteristics and living styles. The derivative spatial structure has altered street patterns and influenced tourists' experience of space.

### 6.1. Structural Composition of Old Street Façades

We analyzed space composition via direct projection, angular cuts, and unfolds. The entire region was divided into five sections by referring to existing street names. The projected elevations of all sections (Figure 5) were categorized by shops with repetitive and rhythmic sun shades (street Section 1) or highly elevated chimneys, gables, shopping malls, or apartments (street Section 2). Section 5 is an elevated connecting trail and skywalk that creates an interesting approach view to the entrance of Old Street.

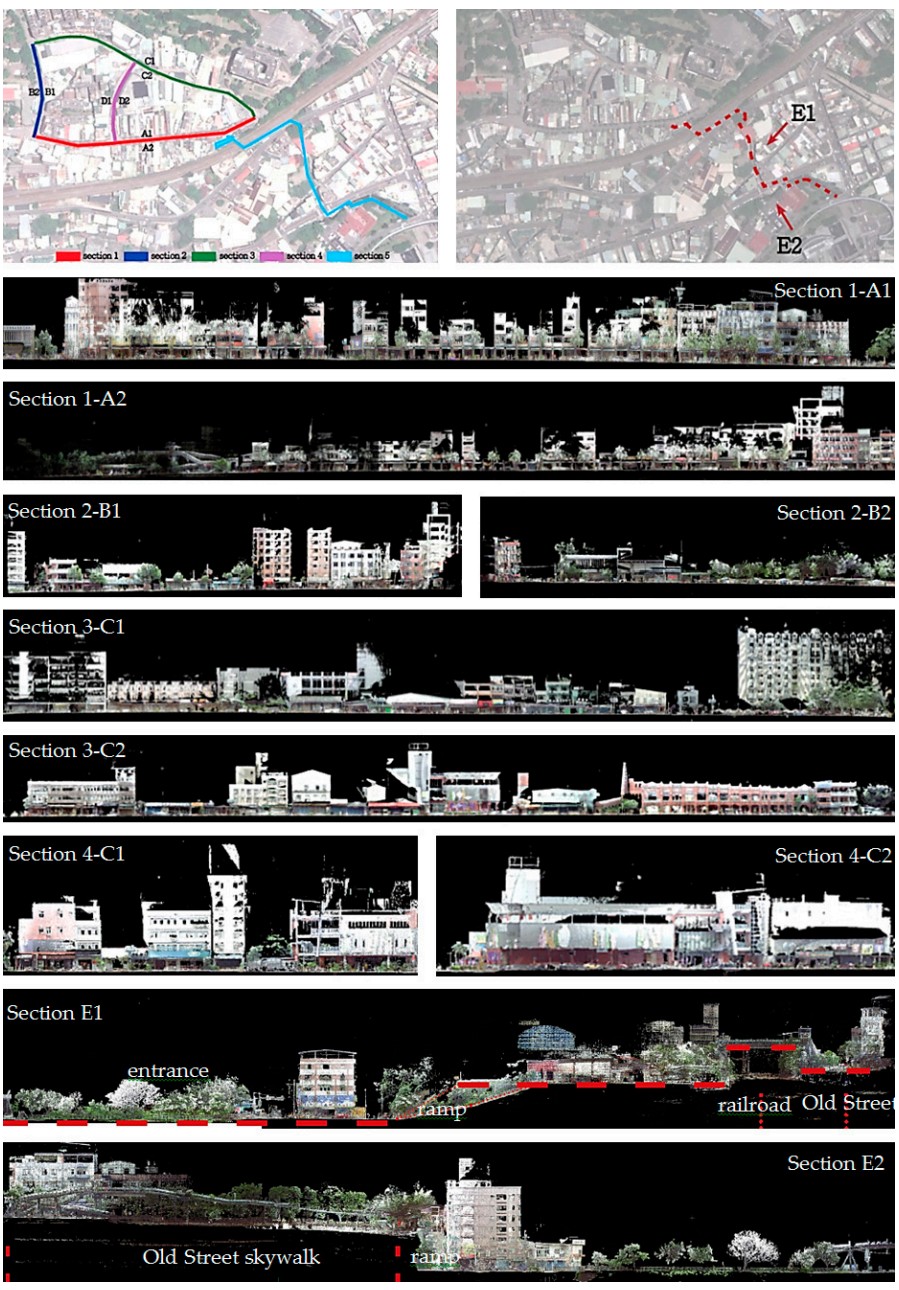

**Figure 5.** Elevation of Old Street and connecting paths.

We used a special approach with an angular cut through the entire town (Figure 6) to continuously illustrate the themes of Old Street through: (1) Connections between trails, street, railroad, and skywalk, and (2) the connections between sightseeing locations and uniquely characteristic buildings.

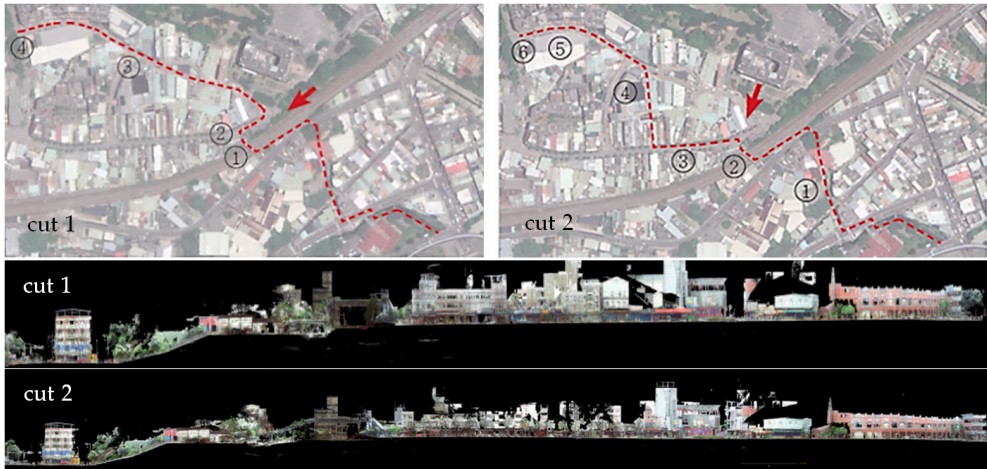

**Figure 6.** Two angular cut sections through Old Street.

The third approach of creating sections was applied to follow the tourists' paths (Figure 7). All the sections illustrate integrated images and interrelationships between the space, structure, terrain, and local characteristics. This approach assisted us to clarify the new symbolic façade expressions with rich variations created by commercial and residence spaces. The unfolding approach enabled the thorough and intuitive inspection of the traditional Old Street structure.

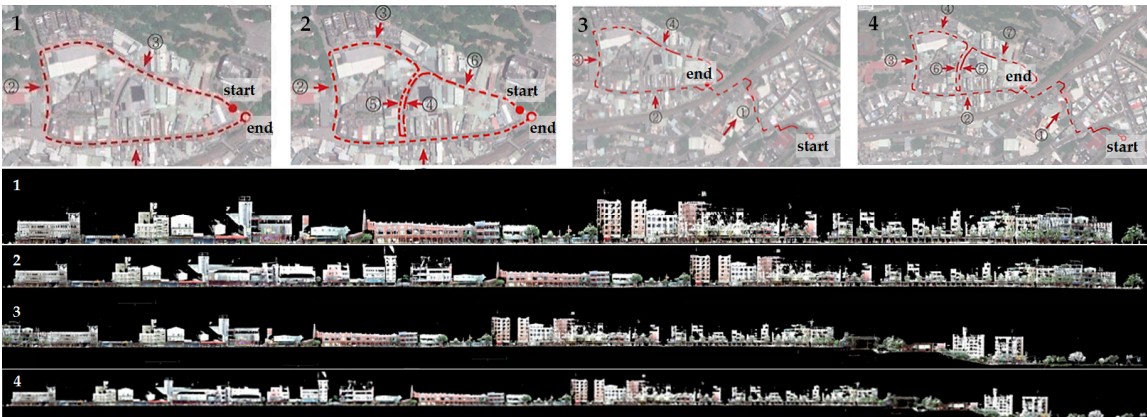

**Figure 7.** Four unfold cuts through Old Street in four routes (from left to right).

### 6.2. Overlay of Old Street Sections

We performed a fundamental analysis on the spatial structure of Old Street. Since curved streets are full of complicated subjects, the retrieval of contour geometries should provide more accurate measurements of dimensions and proportions than photographs or field surveys. The most straightforward approach is to overlay the street from a sectional perspective to identify the special enclosed configuration of open street space and geometries in terms of types, characteristics, allocations, and level of extrusions. In addition to the entire cloud model, each street was divided into 1.5 m thicknesses at 4-m intervals. All the sections were aligned by street centerlines. The overlays not only summarize pedestrians' perspective during the walking experience, but also provide a full scale (1:1) definition of each type (Figure 8).

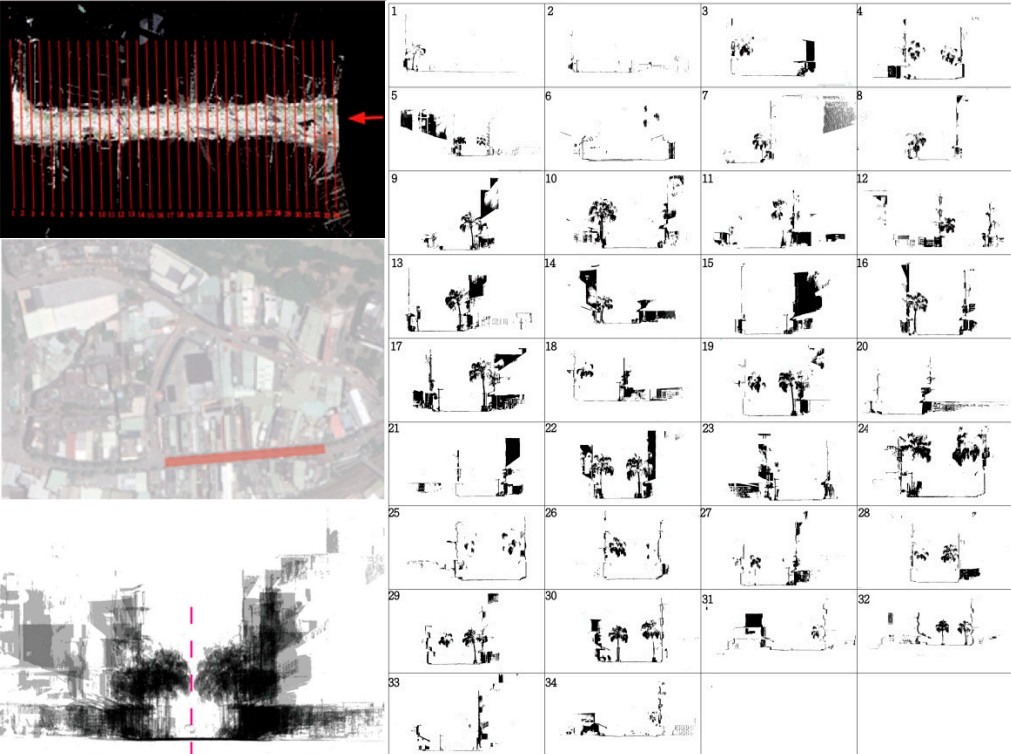

**Figure 8.** The process and 34 outcomes of overlays on street Section 1.

The overlays facilitated the discovery of characteristic and representative vocabularies as different types of responses for each stage of the transformation of street blocks and skylines due to cultural and urban evolution. The iconic vocabularies include railroads, alee trees, skywalks, and pedestrian arcades recessed from shop fronts. The building enclosures and vocabularies have created shapes that are similar to silhouettes of the letter W, sandglass, the letter U, squares, and an upside-down letter T (Figure 9). The shapes were extended to different varieties resulting from the combination of vocabularies and the construction of new buildings with heights and recessed fronts on the ground floor on both sides. Diversity is also determined by the landscape, contributed by former promotion projects, which defines a softened boundary on the sideways and skylines.

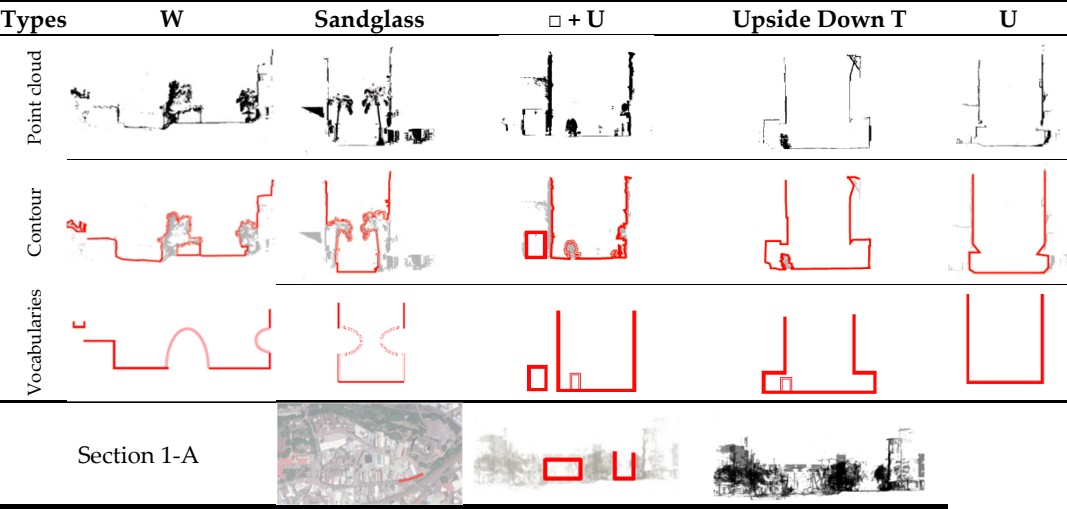

**Figure 9.** *Cont.*

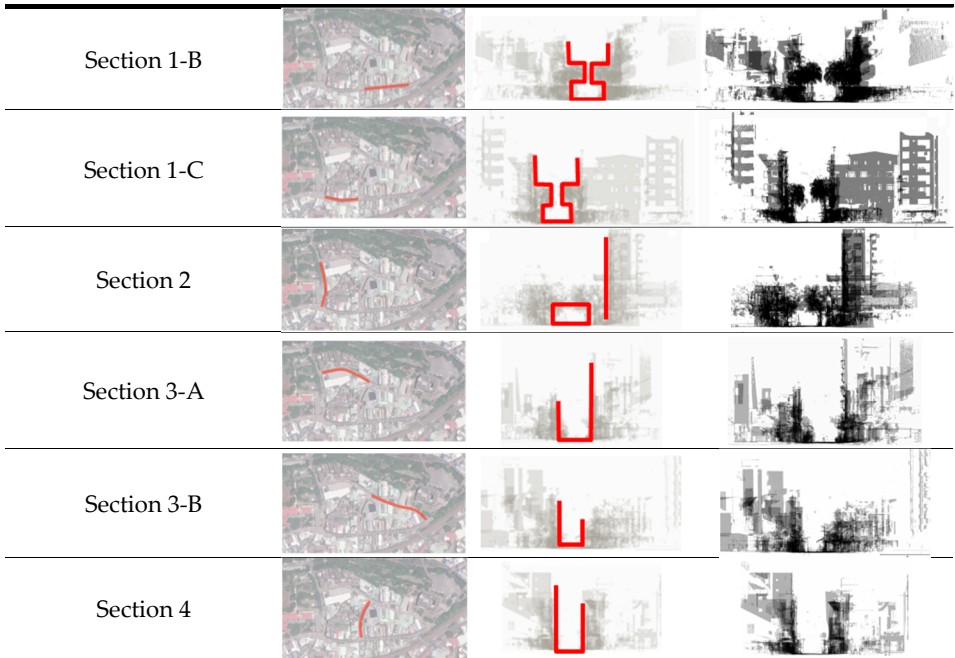

| | | | |
|---|---|---|---|
| Section 1-B | | | |
| Section 1-C | | | |
| Section 2 | | | |
| Section 3-A | | | |
| Section 3-B | | | |
| Section 4 | | | |

**Figure 9.** Characteristic vocabularies of sections and major types of each street section.

### 6.3. The Second Contour

Commercial activities and residential needs contributed to the creation of the second contour and the characteristic street space of Yingge's Old Street. Due to daily living needs, local residents remodeled building façades to install commercial or air-conditioning facilities. Six types can be identified (Figure 10) based on the scale and the construction methods used. The first type, which essentially fulfills daily life needs, includes balconies and steel window grating. The second, third, and fourth types represent three ratios of façade, which are used to install advertising panels or shop signs on the limited space available on building fronts to attain a competitive advantage over other shops on the same street. The fifth type constitutes extruded panels. The sixth type includes sheet metal for additional space needs on the roof level. Based on the observed distribution patterns, Old Street is replete with various types of shop renovations that contribute to the diversified second contour appearance, especially in Section 1.

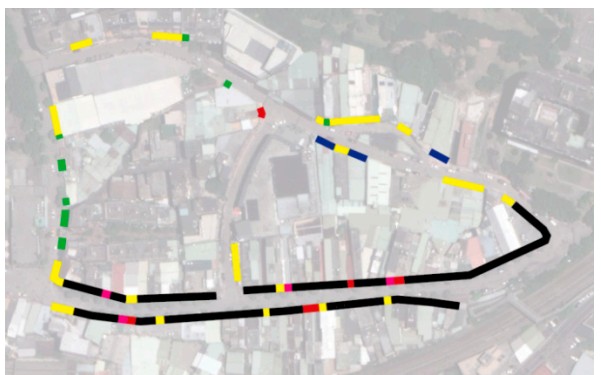
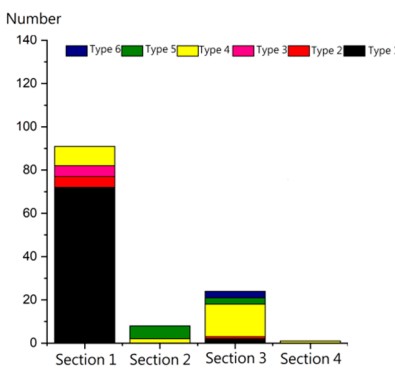

**Figure 10.** *Cont*.

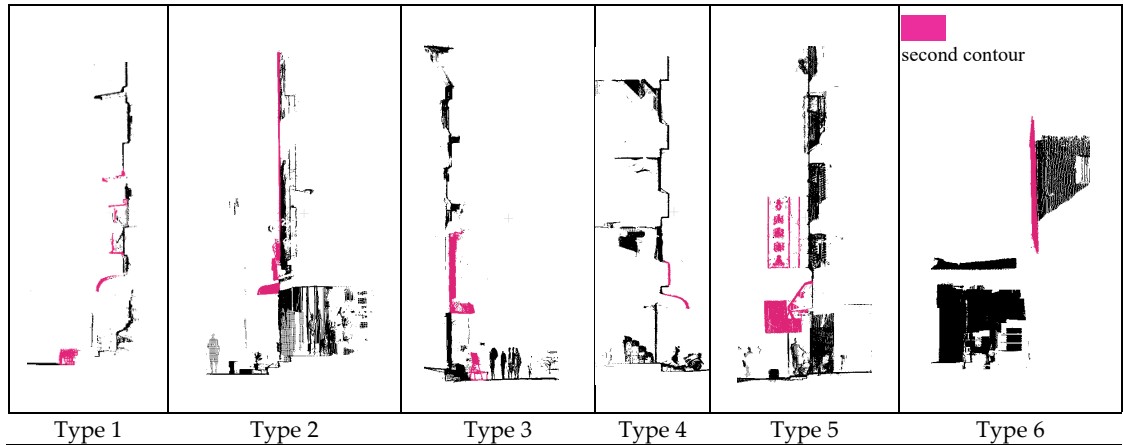

**Figure 10.** (Top) The distribution of second contour on each street section and (bottom) the six local types of the second contour.

The second contour exhibits highly dynamic vitality in terms of patterns and creativities that significantly contribute to local cultural identities. The Old Street second contour originated from the mixed use of residential and commercial spaces. Residential floors are located above commercial areas and contain panels, booths, sun shades, or landscapes. The second contour includes both fixed elements and temporary installations. The former mainly originated from the remodeled parts from the Reputable Commercial District Project. The design determined the current consistent appearance of the commercial district. The temporary installations were derived from local festival activities or merchandise displays, which are allocated in different time periods and alter the configuration of street space through lanterns during the Lantern Festival or during business hours. The commercial activities and associated installations on the ground floor change pedestrians' perceptions of the skyline, in which the real skyline is modified and suppressed by the second contour elements and results in a markedly different profile.

*6.4. Commercial Space in Old Street*

Our field survey revealed that the shop fronts can be categorized into concealed, open, and semi-open types of entrances, with merchandize displayed stacked, in booths, on display racks, or a combination thereof. The type of merchandise is closely related to the pattern of display. The combination of both contributes to the dynamics of the street space and tourist interactions (Figure 11). Based on the survey, most shops choose a closed shop front. However, open and semi-open types of shop fronts enrich the street space of Yingge with various displays that function as tourist attractions.

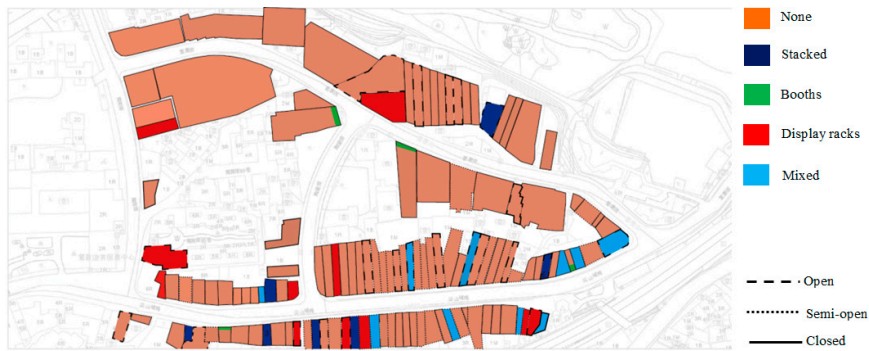

**Figure 11.** *Cont.*

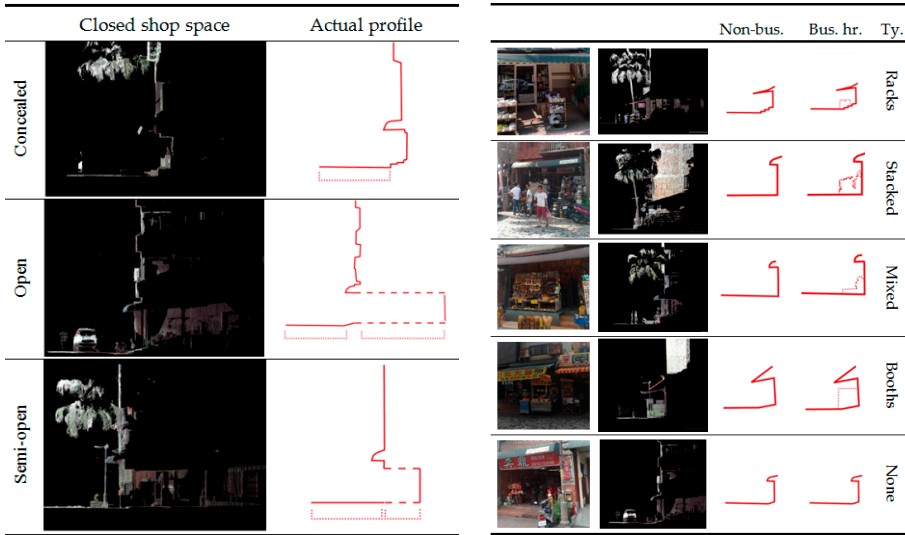

**Figure 11.** The distribution of shop types and display patterns in Old Street (top), and the different types of shop fronts (bottom left) and merchandise displays (bottom right) in business and non-business hours.

*6.5. Horizontal Street Section Analysis of the Open Space Structure in the Hidden Fabric*

Commercial spaces and street open spaces are mutually influential. To illustrate this influence, an overview of the horizontal section was conducted to remove the part of buildings above the second floor. In contrast to the projected solid building property or boundary lines shown in satellite images or cadastral maps, the point cloud model revealed that an open shop front can extend open space sideways from the street and create links between exteriors and interiors into a flexible and sometimes hidden spatial structure of fabric (Figure 12).

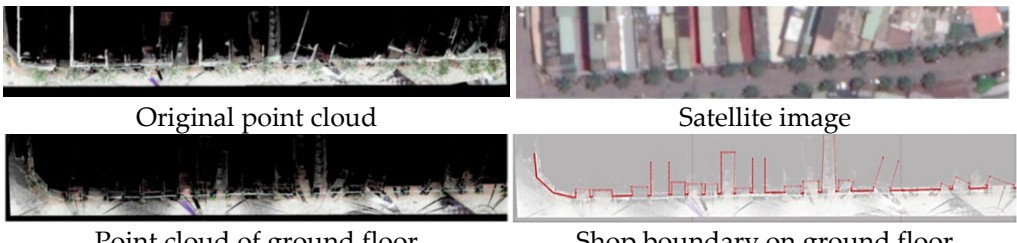

**Figure 12.** Cloud and trimmed cloud (top left and bottom left), a satellite image (top right), and hidden fabric on the ground floor (bottom right).

The point cloud enabled the identification of a dramatic difference in spatial structure between business and non-business hours (Figure 13). The connection between exterior and interior spaces are usually defined by steel rolling shutters during non-business hours. The difference in the area of these spaces is 18.3% (Figure 14), which shows the prevailing types and numbers of open and semi-open shop fronts, particularly in Section 1. This difference also occurs in Section 2 at 15.6%, where recessed pedestrian arcades exist. The average difference is 11.3%.

In general, Section 1 was the first part of town enrolled in the remodeling project of the reputable commercial district. This opportunity attracted many shops with a wide variety of patterns in the spaces. The former intensive bungalow layout was remodeled into a series of concentrated business spaces. Section 2, which features numerous shopping malls and apartments built since 2003, has recessed pedestrian arcades and open spaces designed for public access and leisure needs. This section was full of old large-scale ceramic factories. The difference in the space contour during non-business hours is less significant, given the presence of shops with less open or semi-open access. A significant difference was also identified between satellite images and cloud sections of former periods and

non-business hours in Section 4, due to the new construction of an art center with a large recessed arcade on the ground floor.

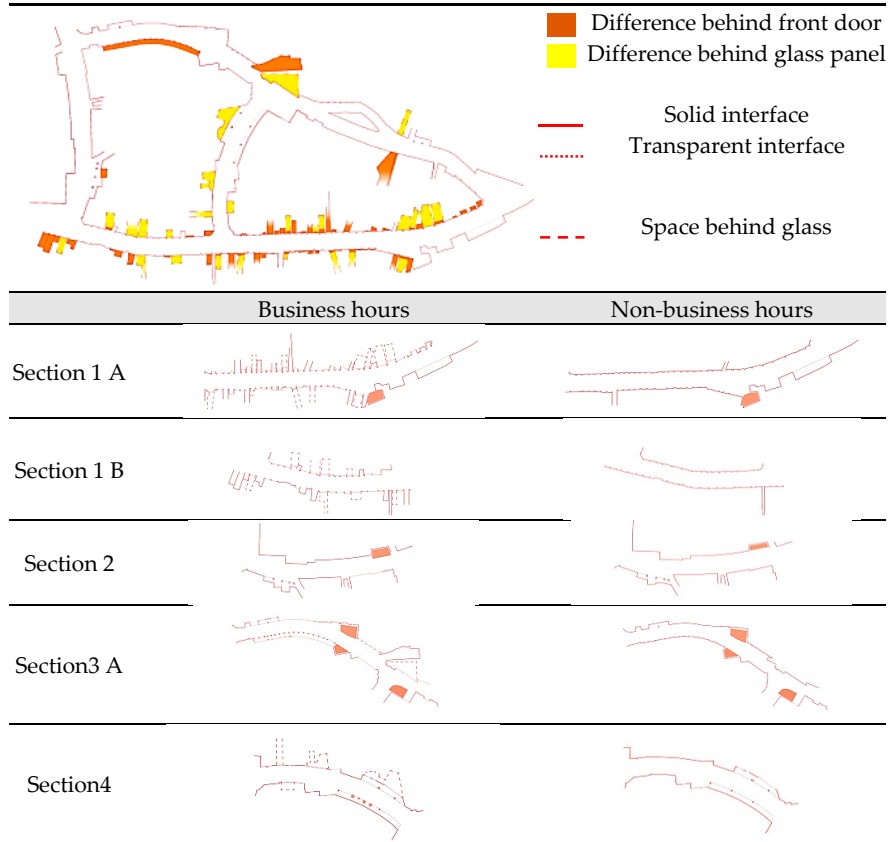

**Figure 13.** The variations in spatial structure between business and non-business hours.

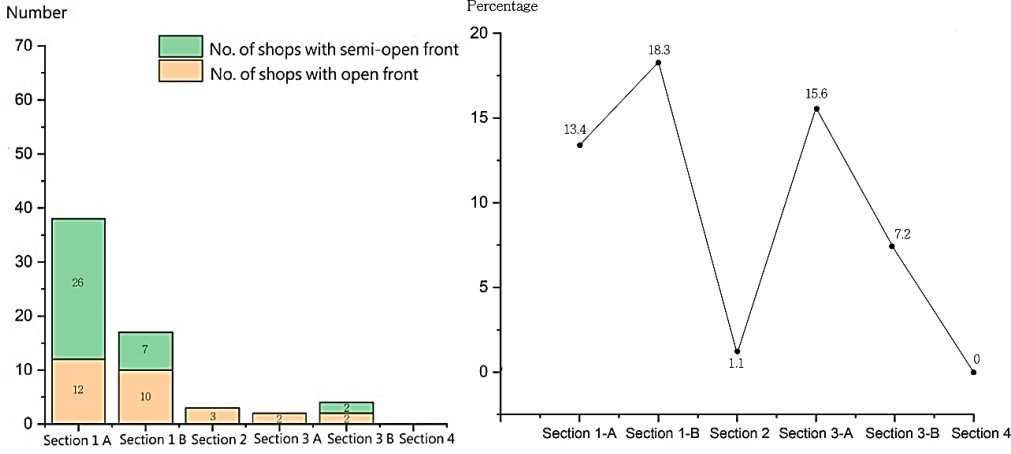

**Figure 14.** The number of shops and area difference between business hours and non-business hours.

In summary, the above-mentioned differences were determined by the sale patterns of ceramics and recessed arcades, or open spaces, due to the new constructions. The renewed business design paradigm, whose transformation originated from the Reputable Commercial District Project, led to the modification of contours, followed by the creation of new characteristics for Old Street.

*6.6. Substantiated Perception of Skylines*

A substantiated perception of skylines, or boundaries, represents the real visual or perceptional boundary of an environment. Although section overlays and projected street elevations assist with defining a boundary, real perception can be limited by tree crowns at a lower altitude. By referring to the real skyline, a substantiated perception of the skyline was drawn (Figure 15).

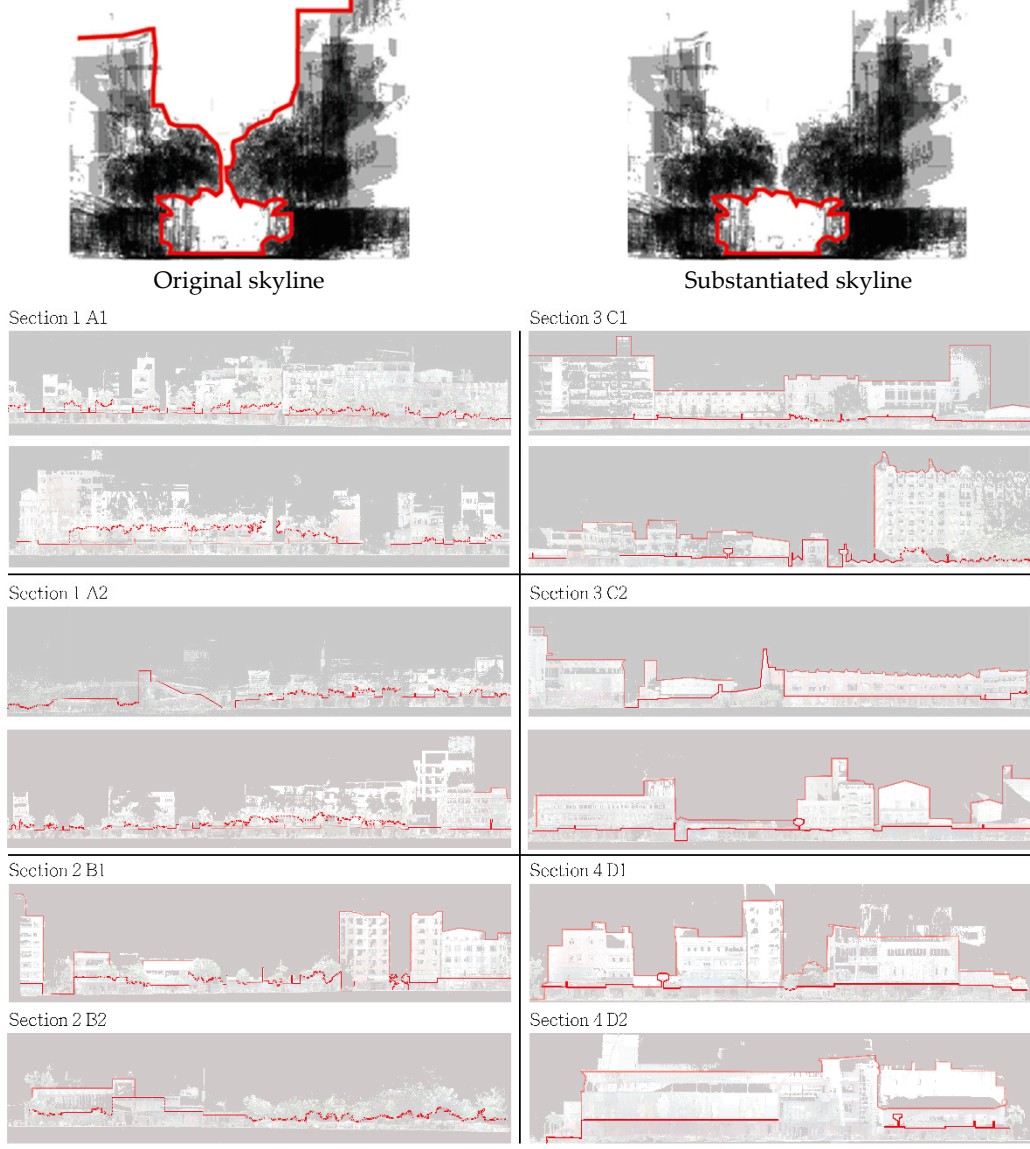

**Figure 15.** The difference in the original skyline or boundary (top left) and the perception of the substantiated skyline or boundary (top right) in each street section (bottom), with different line weights.

The substantiated perception of the skyline varies according to the unique spatial structure in each street section (Figure 15, bottom). For example, the uniform alee trees and sun shades, which created the representative visual image of Section 1, define the perceptional boundary. Section 3 inherits a relatively smooth skyline with less deviation in altitude due to the lower tree crown. The diversified building styles and elements with a lack of alee trees in Sections 2 and 4 create layers of skyline in different depths. Overall, the variation in each section's space structure can be observed through the substantiated skyline, further illustrating the integrity of the original planning strategy (Figure 16 top) in a contrastive fashion.

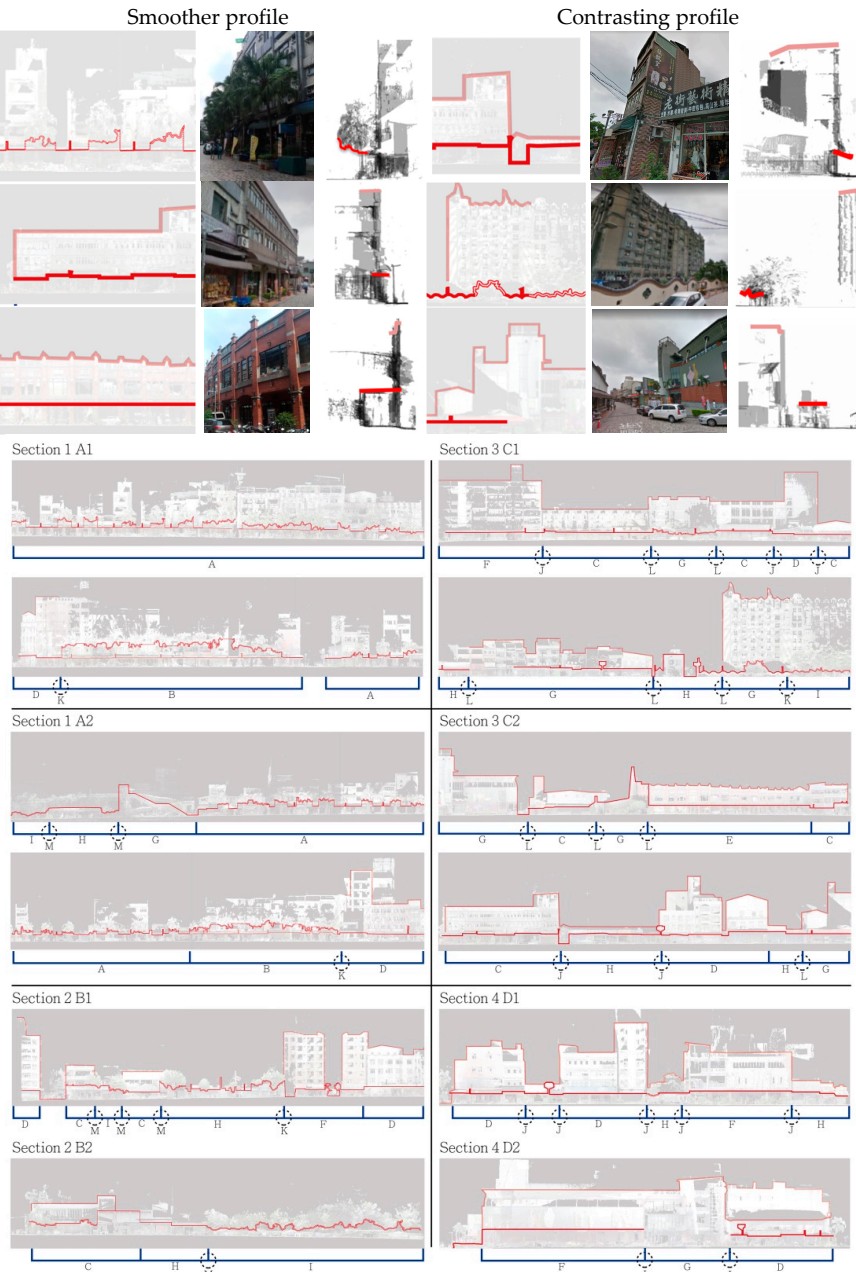

**Figure 16.** The schematic classification of substantiated skyline types in a smoother or contrasting profile (top), and the types or breaks in each street section (bottom).

The remodeling of each street section was completed under different schedules or over various periods of time. The project-defined items, which were not consistently designed, contribute to the diversities of patterns (Figure 16, bottom). The inconsistency of the second contour, such as the alee trees or the allocated sun shades, created street-based experiences with versatile perception patterns of the substantiated skyline. In other words, government promotion projects are closely associated with the layered composition of the second contour. The substantiated perception of the skyline mirrors the planning integrity of the street. As a result, the government planning strategy directly influenced how the local identity developed.

## 7. The Evolution of Old Street Skyline Profiles over Decades

The 3D-scanned cloud model was combined with historical cadastral data to define decade-based skyline profiles. The purpose was to elucidate chronological changes by recovering the old skyline to

illustrate the changes that occurred during different time periods. The data included as-built scans from 2018, as well as drawings derived from the story numbers indicated on cadastral maps from 2005 and 1970. Figure 17 illustrates the changes that occurred in each street section.

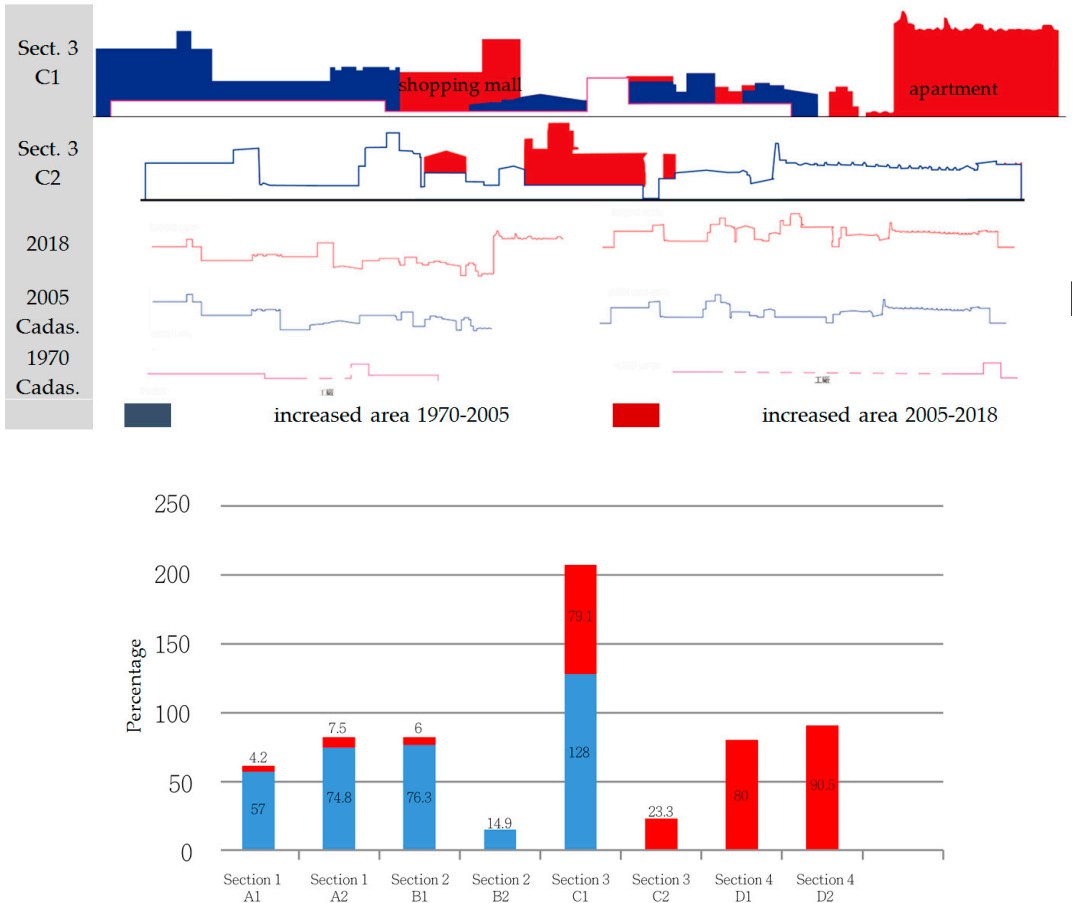

**Figure 17.** The chronological changes of Old Street skyline profiles in street Section 3 based on cadastral maps (top), the increased percentage of projected area in each street section, and the chart of increased amounts (bottom).

To understand the level of changes, the increased area was estimated between 1970 and 2005 (Figure 17, bottom, blue) and 2005 and 2018 (Figure 17, bottom, red). The first time period reveals that most of the elevation has changed. The renovated area increased 50–80% due to the improvement of the economy in street Sections 1 and 3. Section 3 increased 207%, in which the industrial transformation featured many old factories that were remodeled into apartments. Most of the street underwent a significant magnitude of change. A number of new construction projects occurred between 2005 and 2018 in Section 3 with 79.1% and 23.3% increases in shopping malls and apartments, respectively; Section 4 with 80% and 90.5% increases in shopping malls and art centers, respectively; and Section 1 A2 with a 7.5% increase in hotels. Some of the increases were due to illegal construction that was added to existing buildings.

The development of the Old Street revealed a sustainable development process that was subject to economic recovery and government promotion projects. Remodeling in the 1970s changed the townscape of factories and bungalows. The industry transformation, starting in 1990, and the government promotion, starting in 1997, dramatically reshaped the skyline twice. The spatial structure is constantly evolving, although some part of the skyline may be created out of context. With limited concern about the environment, the development process now involves creating spaces for more

sustainable programming of local ceramic culture in promotion, education, or exhibition, by means of galleries, DIY workshops, or classrooms.

Instead of traditional photos and documentation, we assessed the analysis of the skyline profile from 3D as-built street scans and cadastral maps. Although scans of the old townscape do not exist, the skyline of early days was estimated from the story numbers indicated on local cadastral maps. With an additional reference to boundaries in satellite images, a new chronological record of changes could be created to illustrate the evolving process of urban fabric in a broader hierarchy by co-relating to the developing footage on maps.

### 7.1. Chronological Changes in Street Sections

With the reference of historical photos and documentation, we subtracted part of the point cloud model to recover the configuration of the old space. New buildings after promotion projects continued reshaping the contour and profile of Old Street. This subtraction method revealed that demolition and construction most directly influenced the skyline. The new construction of chimneys created a local highlight. In contrast, some restorations of building enclosures had little influence. This evolution involved reinstallation and reminiscence of the old local identity, even though its primary purpose was to improve the profits of local businesses. With the assistance of the cloud editing function, most of the new buildings or facilities were removed to enable a real-scale comparison (Figure 18). However, the process was limited by one-way editing without the reinstallation of former demolished parts. Although the comparison applied cadastral data for estimation purposes, the correlation of as-built detailed geometries with cadastral data produced archived data that evolved with opened-up access of interpretation.

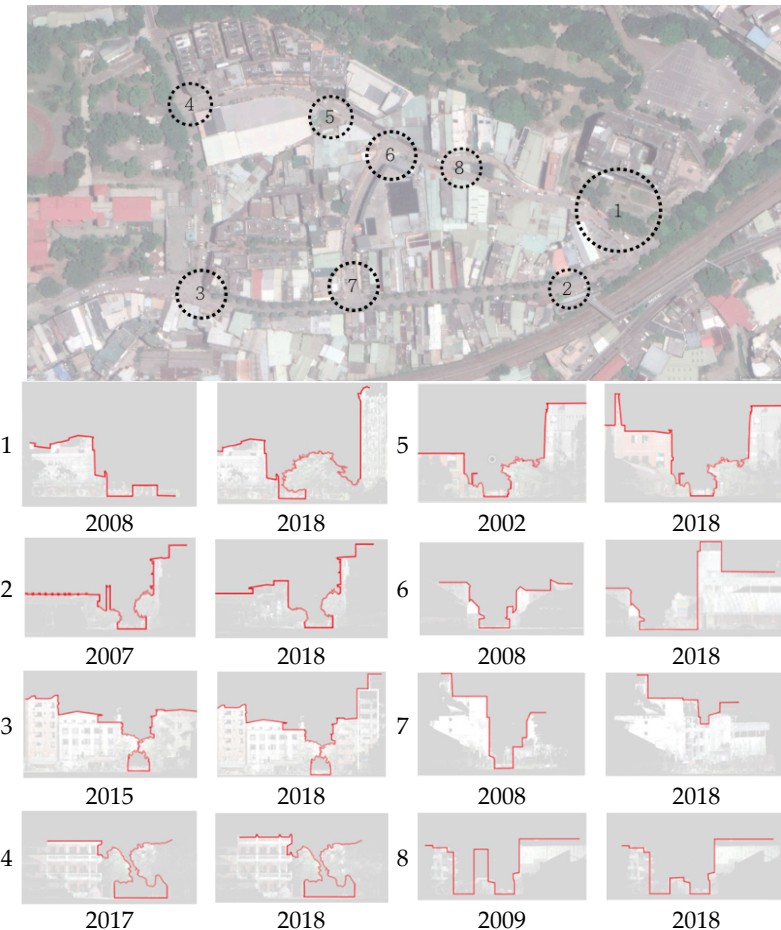

**Figure 18.** Chronological changes of street sections in eight important sightseeing locations.

### 7.2. Three Types of Interdependence in Space Developments

Creative ceramic activities or classes are designed in association with different levels of the reuse, remodeling, or reconstruction of old spaces. Since government promotion was mainly limited to infrastructure, the local development efforts show different types or interpretations of interdependence between the three dimensions of sustainable development in terms of original old building spaces, remodeled spaces, and newly constructed spaces.

### 7.2.1. Original Old Building Spaces

An old kiln was reused as a ceramic DIY workshop with a refurbished building enclosure. Other than kiln setting, many shops on Old Street also offer similar DIY experiences.

### 7.2.2. Remodeled Spaces

Pedestrian arcades were added as a physical buffer between the shops and the street, or a soft interface reached out between the local residents and tourists. The arcade is sometimes used as DIY space.

### 7.2.3. Newly Constructed Spaces

New buildings were designed for better programming of space efficiency for ceramic-related activities. In addition to the government initiated ceramic museum, a new private gallery was initiated as a multi-functioned gallery for education, exhibition, lectures, workshops for local artists, and creative artworks. The museum or gallery represents a new image in a new era. New apartments with better designed living conditions were built for local people and those working in nearby cities.

## 8. AR for Contour Interaction

AR for the interaction of contour display was created as an integrated system of satellite images and point cloud data. The flowchart of the AR application is presented in Figure 19. We created 28 images for front and back display in Photoshop® (Adobe Inc., San Jose, California, USA), using seven sets of overlaid cloud images, and all was exported to SketchUp® (Google Inc., Mountain View, California, USA) to create models. The app was created in Unity® (Unity Technologies, San Francisco, California, USA) using ARKit SDK (Apple Inc., Cupertino, California, USA).

The interacted items and system interface of the app included the design of buttons, satellite images, and section overlays.

As shown in Figure 20, buttons were provided on both sides of the screen: Buttons on the left select satellite images, and buttons on the right select section overlays. All the contents can be selected individually or cumulatively. Pressing the "New" button will bring up Yingge as-built section overlays, indicated by the buttons numbered from one to seven.

For the section overlay, old and new street sections can be selected and displayed. As shown in Figure 20 (bottom left), pressing the "Section 5" button will bring up that section to illustrate the aligned overlaid result of the street sections.

As shown in Figure 20 (bottom right), satellite images can be projected on a table using one or multiple buttons to display the image for the desired year to facilitate stage-wise inspection of the development of the urban fabric.

A satellite images and overlaid images can be combined by displaying Old Street images above satellite images or cadastral maps (Figure 20, top and middle).

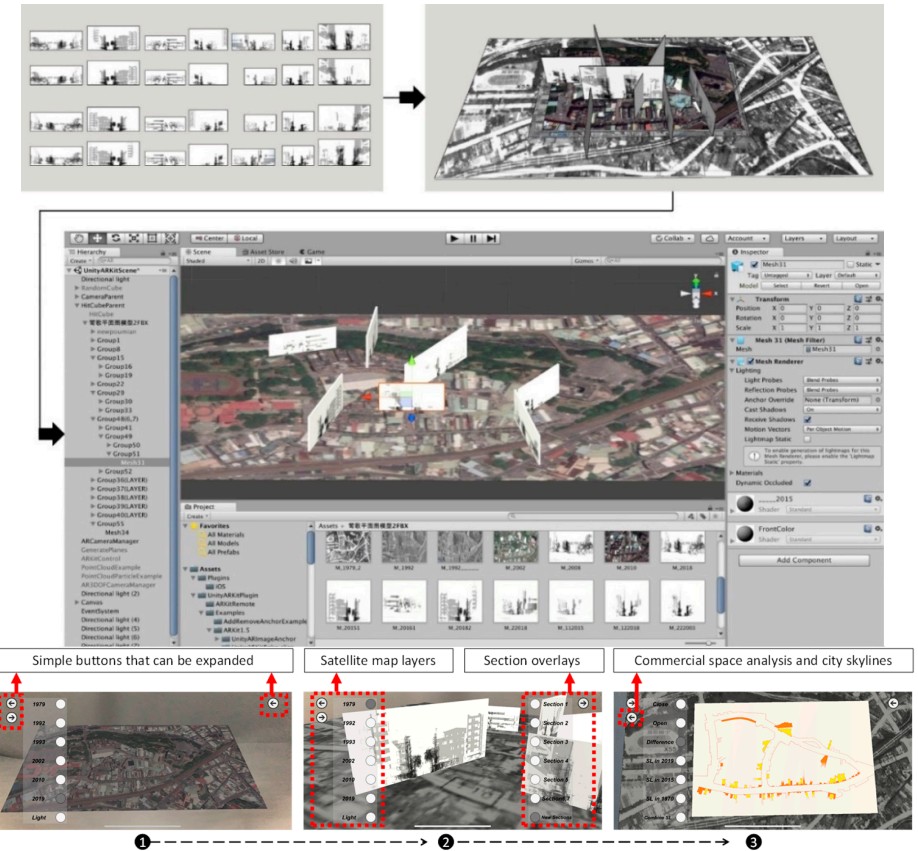

**Figure 19.** Augmented reality (AR) application creation diagram (top) and operational interface (bottom).

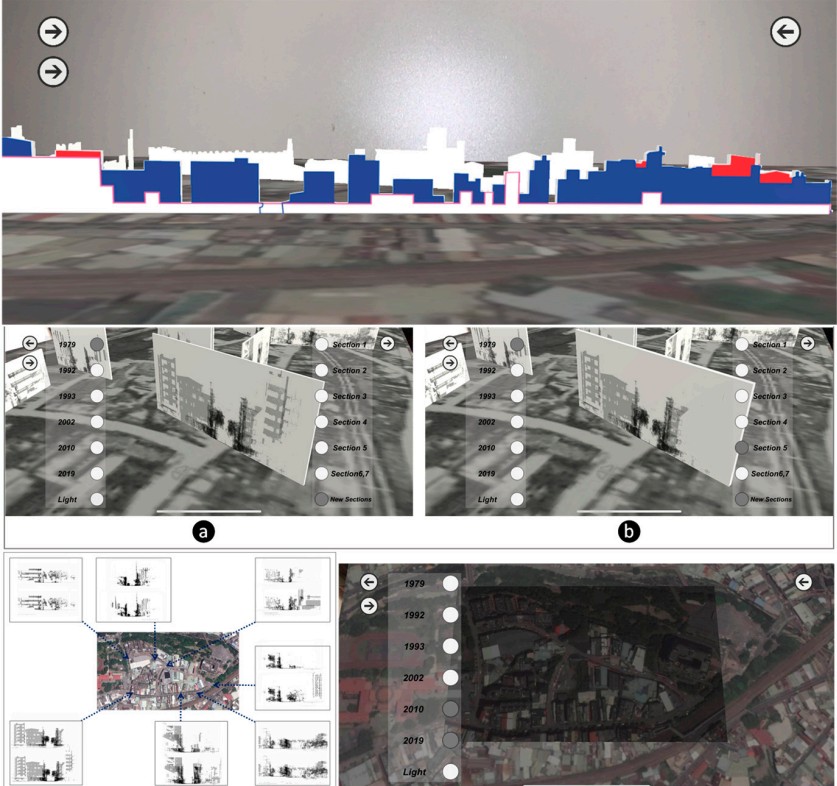

**Figure 20.** Screenshots of section overlay over satellite images with chronological changes (top and middle) and satellite images of different eras (bottom).

## 9. Discussion

The inspection, with the entire Old Street façade revealed, facilitates understanding from both micro and macro perspectives. The section across the entire Old Street region differentiated the weight distribution of development. While a comparison was made by year, diagrams placed side by side showed the trail of the development sequence in the region. Accordingly, an analysis was performed from a macro to micro scale, in terms from satellite images, plans, skylines, facades, and sections.

### 9.1. Large-Scale Shopping Mall or Important Window to Cultural Tourism

Commercial patterns usually mirror local cultural identity. Even souvenirs, which are often used to promote local identity, represent formulated subjects that originate from a long-term evolution of local culture. All of Yingge is a large-scale shopping mall of ceramic-related objects and cultural experiences. Compared to the inspection of individual characteristics of a shop or a street from the bottom up, a whole study was conducted from a macro perspective integrated with the top-down process.

Yingge is also an important window into cultural tourism. Part of the footage in cultural development is determined by the substantiated subject of representatives. The two-way inspection between macro and micro views confirms the role and development process in local cultural evolution. Consequently, all the overlays that contrast the differences in multiple dimensions verify the configuration and distribution of changes from macro and micro perspectives, or, for example, the changes to the urban fabric from a macro view, or the remodeling that occurred to the plans, elevations, or skylines from a micro perspective.

### 9.2. The Second Contour

The second contour of Old Street was programmed, designed, and evolved with local characteristics under the goal of creative management of commercial space. The shop signs, sun shades, and alee trees created the symbolic structure of the contour and enhanced the walking experience under the substantiated skylines. The modified boundary of perception produced a distinct commercial image that extends local ceramic culture from the manufacturing industry to sustainable tourism.

The identity of Old Street is mainly determined by local cultural characteristics. A clear definition of those characteristics can impress people, aiding the long-term development of tourism. The ceramic image clearly defines the local identity, and is further enhanced by the second contour originated from the patterns of sales, shop fronts, and displays.

The consensus and responsibility of the government and local residents created a type of constraint to maintaining and protecting the old identity. As a result, the development of the second contour followed the consensus for a characteristic interpretation that can be temporary, but flexible or adaptable enough, for existing fabrics. For example, an atmosphere is usually created during the Lantern Festival with a new substantiated skyline defined by red lanterns. The consistent cultural characteristics of the second contour have attracted tourists by establishing a long-lasting tourism memory. In other words, the sustainability of local tourism can be created by the contour or fabric setting that has a flexible yet appropriate interpretation of identity on different occasions.

Not only is the entire town engaged in some form of selling, but each store operates in the same way by using many forms of media to promote individual characteristics through the second contour. This contour was one of the media modalities used and continually modified during the three stages of development. The collective images of each street or the even the entire Old Street continue to create an integrated image that contributes to the transformation from industry to cultural tourism. Although the vocabularies may not always be consistent, the process constitutes a consistent pace of evolution.

### 9.3. Joint Identity with Nearby Cities

In addition to the daily-use bathrooms and kitchenware, Yingge has created a number of ceramic artists dedicated to the preservation or creation of traditional art in modern times, with hands-on

experiences provided by museums, community classrooms, and workshops. Old Street has joined the surrounding ceramic museum, parks, and workshops to form an enlarged cultural region of identity. A skywalk was also installed to connect Old Street and the ceramic museum in 2009 during the process of the second stage fulfillment of the Central Pedestrian Zone Project. The connection between the two main popular locations was considered successful in promoting sustainable tourism and integrating commercial activity and exhibitions.

Multiple shop types, openness, and front displays have diversified correspondence with the old cultural identity in a new era. The cultural diversity has become incorporated with the identity of nearby cities to form a larger integrated cultural neighborhood, which not only enriches the characteristics of each city, but enhances tourists' comprehension of regional development. The differentiation and integration of local culture has augmented the Old Street image and simultaneously illustrated the importance of past promotion in cultural infrastructure and economy.

### 9.4. Old Street Sustainability

Planning integrity and the incorporation of the local government are critical factors in Old Street's sustainable development. The project for Old Street commercial remodeling successfully transformed the local manufacturing industry into ceramic culture tourism. This sustainable effort was extended to project-defined street elements with the clearly defined identification of the second contour, and successfully contributed to the important tourists' views of the street. The substantiated skyline also constitutes a symbolic identity with a modified street fabric. The project-defined elements, such as plants and shades, at each street section are not consistent with each other and lead to a sudden disruption of the skyline. This inconsistency resulted from individual street section-dependent attempts to fulfill function needs.

To sustainably preserve heritage, an appropriate strategy must be developed to redeem unused space with novel meanings and roles. This development must also consider local cultural identity and industry characteristics with the involvement of the government and the commitment of local residents. Yingge's industry transformation has altered old business types by combining ceramic culture and the Reputable Commercial District. Sustainability in tourism can be represented by the changes in urban fabric, as shown in the structure and configuration of the second contour and skyline.

The evolving configuration of the skyline can be considered evidence of changes that occurred during different urban fabric development stages. The transformation of an industrial city is part of the sustainable development process that was achieved through the demolition and reconstruction of buildings. Changing low bungalows and factories to diversified configurations with higher altitudes involves a long-term effort. The remodeling efforts or reconstructions were mainly initiated by the owners who, at a certain point in time, were motivated by the promotions and considered possibly being involved in a program of ceramic-related activities. The detailed displacement of original factory workers was untraceable. However, considering the ceramic history and business scale in early days, the majority of the workers were temporary employees from local areas. The displacement of the workers due to the shutdown of factories might be limited based on their own residence, but worse in the long term since the recession started.

The new skyline, comprised of chimneys and buildings with gables, which originated from reminiscent images of the old townscape, can be recognized as a clear illustration of the commitment to creative tourism with the sustainable awareness of local culture and identity. Although the new construction may not be considered a perfect process of evolution, and the factories can be remodeled for reuse for creative culture products, the local cultural image was applied during the design of building elements, such as façades and shop fronts. The designated effort and emphasis resulted in different approaches and outcomes. Eventually, the remodeling regained the recognition of tourists, and the transformation was ultimately successful.

The key factors in the Old Street development included existing space redevelopment, resident involvement, characteristic categorization, planning integrity, and government promotion. Most

of the factors are related to programming of recreated value through the redevelopment of space, reconstruction of buildings, and the restructure of urban fabric. For example, a local shop was renovated into a new Starbucks, with a second contour composed of old building vocabularies. The combination of cultural elements and a major coffee shop chain created a successful representation of local image and achieved a high level of exposure and popularity among new generations. The old factories, which were redeveloped into shopping malls offering merchandise and local designers' products, are typical instances of industry transformation during the creative development of local culture and identity, in terms of an environment removed or improved, and the space suitable for social interactions.

*9.5. Creativity or Sustainability*

How the intensive changes to the urban fabric meet the specific needs of local culture remains to be seen. However, the highly polluted kilns in the old days were removed to eliminate the concerns about environmental protection and provide space for more versatile social development. The remodeling or reconstruction, which is usually considered a negative indicator of the environmental dimension in sustainable development, was completed in several patterns that connect to cultural tourism. The local development efforts have presented different types of interdependence between the three dimensions in sustainable development in terms of old buildings, remodeled spaces, and newly constructed spaces.

In general, the ceramic-related culture activities on Old Street include DIY workshops, exhibitions, and the Ceramic God Festival. The former requires space. The latter is held yearly on Old Street or open space nearby. Fortunately, cultural activities, which occur in a space with a temporary deployment of fabric, have been running for years as some of the most sustainable social events. So why do we need remodeling or reconstruction when the cultural sustainability can be achieved outdoors?

Creative tourism could offer development opportunities for smaller communities [35]. The promotion of ceramic culture involves many aspects that originated from traditional culture, legends, religion, or fabric. The social development between the local people and tourists occurs in many types of interactions; e.g., cultural exhibition, education, or entertainment. For example, the Ceramic God Festival is held with a parade or lanterns along Old Street. The festival attracts many tourists every year. No permanent constructions are created. The temporary parade or festival installations positively fulfill the sustainability in creative tourism. Can this festival activity be held in any location without the Old Street fabric setting? The answer is no. The long-term evolvement of the fabric has contributed a significant setting that facilitates the related activities under the background of cultural sustainability. The cherished local identity and history has created a new social dimension that combines ceramic culture in creative industry and business.

## 10. Conclusions

A purposely selected display of sustainability can be achieved using a database system to enable a typical architectural analysis, including the use of traditional drawing types and maps as a link between the vertical and horizontal hierarchy of temporal models. The scanned point clouds serve as a time-stamp framework that can be used to interpret and collaborate with other data sources in a vertical hierarchy. The point cloud database provides a collaborative environment for data integration and AR interface for versatile layers of related maps in a horizontal hierarchy and multi-purposed pedagogical or tourist navigation in vertical hierarchy. The real world can be analyzed with the support of the point cloud, creating a more effective AR interface. In addition, the point cloud facilitates a qualitative and quantitative analysis the local culture evolution in terms of fabric and the second contour.

The point-cloud-oriented spatial analysis of Old Street was used to illustrate the cultural character of the area and the spatial experience in cultural tourism. The results showed the structural composition and symbolic expressions of façades, the characteristic vocabularies of street sections, the six local types of second contour and their distribution in each street section, the distribution of shop types and display patterns on shop fronts, the open space structure in the hidden fabric between business and

non-business hours, and substantiated perception of skylines. The analysis of the evolving Old Street skyline, profiled over decades, illustrates the chronological changes and the increased percentage of projected area in eight important sightseeing locations. Three types of interdependence in space were developed.

The second contour illustrated dynamic vitality in terms of patterns and creativity that significantly contributed to local cultural identities. Shopping space and open street spaces are mutually influential. A flexible and sometimes hidden spatial fabric structure exists. The commercial activities and associated installations on the ground floor change pedestrians' perceptions of skylines, in which the real skyline is modified and suppressed by the second contour elements into a markedly different profile. The transformation of the Old Street fabric was studied under the influence of government projects, commercial activities, local culture identity, commercial and living patterns, and historical stages of development. 3D scans were used to correlate a series of analyses using vocabularies, and overlaid sections to examine special characteristics and changes. The analyses discovered that the second contour determined the profile of Old Street. Due to the high density of commercial activity, commercial and living elements contributed to the configuration of the second contour through façade decorations, shop signs, product displays, window covers, and balconies. The government reform project for Reputable Commercial District, also programmed the scope and elements to promote historical street images and reactivate cultural identity from the past. The alee trees and unified form of the sun shades also contribute to the configuration of the second contour. The second contour, which may be time dependent, varies its appearance within each day as a response to local commercial routines during business and non-business hours.

The high density commercial activity determines the structure of Old Street. Based on the sections constructed in different orientations, perceptions of street boundaries are defined by the design or planning of shop front openness and product displays on the ground floor. Differences also exist in the substantiated perception of the street skyline, which is determined by the second contour. The multi-layered vertical and horizontal perceptions are strongly related to the new commercial buildings and a series of government promotion projects.

A substantiated perception of skyline mirrors the planning integrity of the street. Consequently, the government planning strategy directly influenced how the local identity developed. The evolution of the Old Street fabric was explored through changes to the skyline and sections. We discovered that combining 3D scan data with documentation and maps is beneficial for identifying referable connections between reality and chronological data. Skylines from various time periods were simulated by trimming the point cloud for partial space recovery. This is feasible for regions experiencing major demolitions and new construction occurring within the same property line by the original owners. This remodeling, due to industrial transformation with the support of government projects, has contributed to the major skyline reconstruction in the past, which is an on-going process. Application of the cloud model is more precise and straightforward than the application of traditional photos and documentation. The point cloud, which should be continuously created for paired comparisons, provides a reference that can be correlated with exploration in vertical and horizontal hierarchies.

The environment dimension in sustainable development remains to be explored. The recovery from polluted factories attracted many remodeling and new construction projects, and, consequently, more investments from local residents or outside enterprises. Many shops along Old Street are rented from local residents. The new shops, with new business models, may gradually diminish the ceramic culture to a more commercially-oriented pattern. Whether the evolving model interacts with the social dimension remains to be seen. Whether the interdependence to the environment will evolve into a new conversation pattern between the owner and the tourist also remains to be explored in the future. The three dimensions in sustainable development will be mutually reinforced into a Yingge-specific pattern. Old Street, which was sustainably transformed using much cleaner economic activities, brought wealth to the community and preserved the traditional ceramic culture. Cultural tourism, as a successful

complimentary economic activity, has been developed with rich programming of spaces and with more environmentally-friendly awareness and technologies.

Overall, the sustainability of local tourism can be represented by contours or fabric settings that have a flexible yet appropriate interpretation of identity on different occasions. Future studies should be conducted using return scans of Old Street as a long-term chronological record that will provide a more detailed interpretation of the evolved footage. Scans of dynamic scenes of festivals should also be captured for a more vivid description of local cultural events and strategies to promote tourism.

**Author Contributions:** Most of the conceptualization, methodology, validation, formal analysis, investigation, resources, 3D scan data curation, visualization, original draft preparation, reviewing and editing, supervision, and project administration were conducted by N.-J.S. Additional methodology, validation, formal analysis, investigation, 3D scan data curation, drawing creation, and visualization were conducted by W.-T.H. AR software was programmed by P.-H.D. AR data curation was conducted by all authors.

**Funding:** This research was funded by the Ministry of Science and Technology of Taiwan, grant number MOST 107-2221-E-011-029-MY3 (the first year).

**Conflicts of Interest:** The authors declare no conflict of interest.

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
