# Peer review of "Point Cloud-Oriented Inspection of Old Street’s Sustainable Transformation from the Ceramic Industry to Cultural Tourism: A Case Study of Yingge, a Ceramic Town in Taiwan"

_sustainability, doi:10.3390/su11174749_

Round 1
Reviewer 1 Report
This paper sought to describe and document the transformations occurred in Yingge, a ceramic town in Taiwan in the past 50 years. In this regard, the paper does it in a successful manner, although the images are not always clear or understandable for an architecture non-expert reader.
The problem of this paper is on the whole argumentation to link this transformation with sustainable tourism.
1- It starts with a controversial citation of Tweed and Sutherland (2007) "The environmental factor is the dominant concern of sustainable development, and is conceived as a constraint on human development". First of all, the organizing principle of sustainability is sustainable development, which according to the United Nations General Assembly, 2005, p. 12 consists of «economic development, social development and environmental protection». Accordingly, the environmental factor is one of the dimensions of sustainable development or human development. Therefore, it is wrong to consider one of the pillars for sustainable development «a constraint on human development».
2- Changes in the urban development given as examples of sustainability in the paper:
«rich 216 variations created by shopping malls and apartments» (page 8, lines 216, 217); «the industrial transformation featured many old factories that were remodeled into apartments» (page 15, lines 369,370); «The development of the Old Street revealed a sustainable development process which was subject to economic recovery and government promotion projects. Remodeling in the 1970s changed the townscape of factories and bungalows. The industry transformation, starting in 1990, and the government promotion, starting in 1997, dramatically reshaped the skyline twice» (page 16, lines 366-369); «demolition and construction had the most direct influence on the skyline» (page 16, line 381); «This evolution constituted reinstallation and reminiscence of old local identity, even though its primary purpose was to improve the profits of local businesses» (page 16, lines 383-384); «Sustainability in tourism can be represented by the changes in urban fabric» (page 20, lines 500-501); «The transformation of an industrial city is part of the sustainable development process that was achieved through the demolition and reconstruction of buildings. The change from low bungalows and factories to diversified configurations with higher altitudes constitutes a long-term effort. The new skyline, comprising chimneys and buildings with gables, which come from reminiscent images of the old townscape, can be recognized as a clear illustration of the commitment to sustainable tourism (page 20, lines 504-509); «a local shop was renovated into a new Starbucks (page 20, line 519); «Indeed, the old factories, which were redeveloped into shopping malls offering merchandise and local designers’ products, constitute typical instances of industry transformation in a sustainable manner» (page 20, lines 522-524).
Argueing that «The change from low bungalows and factories to diversified configurations with higher altitudes» is «a clear illustration of the commitment to sustainable tourism» is very controversial and poses some questions: what happened to the local population of the bungalows and workers of the old factories? Have they been displaced? Have they profited from the transformation? or were they victims of the "progress"? Have the new malls interfered with traditional trade? It seems that there has been a disruption from traditional ways of life and traditional landscapes, which contradicts the idea of sustainable tourism. The transformation from a low-rise building skyline to a high-rise building skyline clearly contradicts the idea of preserving the identity of the places, subjacent to sustainable tourism.
The demolition of «the old factories, which were redeveloped into shopping malls offering merchandise and local designers’ products, constitute typical instances of industry transformation in a sustainable manner» is controversial, especially that that was done «in a sustainable manner». someawhere in the paper you should explain in which perspective you see sustainability and sustainable tourism.
Other shortcomings and suggestions for improvement of the paper
The abstract has 240 words and clearly exceeds the permitted 200 words; Some text must be eliminated; the abstract does not present any objectives; Objectives must be added to the abstract; The goals of the papers are not clearly stated in section 3, entitled Research Goal. The authors just need to make the goals clear for the reader through a better formulation of the sentences. I have highlighted several errors in attached pdf, which means that the text needs to be proofread and corrected by an English native proofreader. Page 10, line 271, try to explain in what consisted the survey field. Very important: my suggestion for the argumentation that the transformation of the Old Street was made on a sustainable manner should stress the fact the old factories caused much pollution and have been replaced by much cleaner economic activities that brought wealth to the community and preserved the traditional ceramic activity, yet with modern, more environmentally friendly technologies and developed cultural tourism as a successful complimentary economic activity. My view is that this paper should link the transformation not so much with sustainable tourism, but with creative tourism and that the authors should read important papers on creative tourism, such as: Greg Richards (2019). Creative tourism: opportunities for smaller places? Tourism & Management Studies, 15(SI), 7-10.

Author Response
Dear Reviewer:
One behalf of my co-authors, your comments are very helpful and enlightening. Would you please check the attached file to view the responses made to your comments.
Your comments are highly appreciated.
Best regards,
Naai-Jung Shih

Reviewer 2 Report
1. paper title is too long. Please reduce.
2. fig. 1. Please label each sub-figure (e.g., a), b), ...). Also add descriptive captions
corresponding to each sub-figure.
3. line 124. Grammar "... The data functions"
4. General comment: throughout the manuscript there are many instances of very long sentences. For example, lines 125-128. This negatively affects readability. Please correct throughout the manuscript.
5. line 183. what are 'D scans'?
6. line 188. correct to "....photogrammetric modeling..."
7. fig 17 bar graph. add legend for red/blue colours.
Author Response

(The authors gave the same response as above.)

Round 2
Reviewer 1 Report
Dear authors,
Thank you for your responses and improvements. Now I agree with the publication of the paper conditioned to a minor improvement that you did not consider and I had asked after the first revision:
«the abstract does not present any objectives; Objectives must be added to the abstract».
You have reduced the abstract to 200 words as required, but did not add any objectives to the abstract. Just add one sentence with the objectives, bearing in mind that it should not exceed the limit.
Kind regards